# A Ka-Band SiGe BiCMOS Quasi-F$^{-1}$ Power Amplifier Using a Parasitic Capacitance Cancellation Technique $^\dagger$

**Vasileios Manouras ***  **and Ioannis Papananos ***

School of Electrical and Computer Engineering, National Technical University of Athens, 15780 Athens, Greece
* Correspondence: vasileiosmanouras@mail.ntua.gr (V.M.); papan@elab.ntua.gr (I.P.)
† This paper is an extended version of our paper published in the 2022 Panhellenic Conference on Electronics & Telecommunications. Manouras, V.; Papananos, I. A Ka-Band Quasi-F−1 Power Amplifier in a 130 nm SiGe BiCMOS Technology, 2022 Panhellenic Conference on Electronics & Telecommunications (PACET), 2022, pp. 1–5, doi:10.1109/PACET56979.2022.9976366.

**Abstract:** This paper deals with the design, analysis, and implementation of a Ka-band, single-stage, quasi-inverse class F power amplifier (PA). A detailed methodology for the evaluation of the active device's output capacitance is described, enabling the designing of a second-harmonically tuned load and resulting in enhanced performance. A simplified model for the extraction of time-domain intrinsic voltage and current waveforms at the output of the main active core is introduced, enforcing the implementation process of the proposed quasi-inverse class F technique. The PA is fabricated in a 130 nm SiGe BiCMOS technology with $f_T/f_{max} = 250/370$ GHz and it is suitable for 5G applications. It achieves 33% peak power-added efficiency ($PAE$), 18.8 dBm saturation output power $P_{sat}$, and 14.7 dB maximum large-signal power gain $G$ at the operating frequency of 38 GHz. The PA's response is also tested under a modulated-signal excitation and simulation results are denoted in this paper. The chip size is $0.605 \times 0.712$ mm$^2$ including all pads.

**Keywords:** inverse class F; quasi-F$^{-1}$; power amplifier; cascode amplifier; harmonic tuned PA; mm-wave PA; Ka-band PA; 37–40 GHz PA

## 1. Introduction

The surge in consumer demand for mobile data, as well as their preference for superior performance, enhanced quality, and increased reliability, has led to a continual rise in the need for elevated data rates and reduced latency [1]. This demand has been a major driver for Federal Communication Committee (FCC) to announce procedures for mm-wave auctions and define the 5G new radio specifications [2]. Ka-band (26.5–40 GHz) is one of the mm-wave bands of interest, since it can support higher data rate communication, lower latency, and smaller equipment size [3]. The final scope behind the efforts of the research community in the Ka-band circuit technologies is to enable various applications, such as space telescope, close-range targeting radars, satellite communications, and fifth-generation enhanced mobile broadband applications (eMBB) [4,5].

The power amplifier (PA) is an essential component in the mm-wave transmitter chain, as it contributes significantly to power consumption and linearity [6]. In 5G wireless systems, the utilization of complex modulated signals that possess a high peak-to-average power ratio (PAPR) necessitates the power amplifiers to sustain high efficiency while operating over a broad range of power at back-off. This requirement arises because the PAs need to operate in such a way that they can accommodate the high variations in signal power that are inherent in signals with high PAPR. This is important in order to ensure that the system can operate with a high level of performance, even under challenging conditions [7]. However, developing highly efficient silicon-based mm-wave PAs that can offer satisfactory PAE even at power back-off (PBO) is a challenging task. The reason behind this challenge is not only the inherent trade-off between break-down voltages

and maximum speed in silicon-based transistors, but also the limited quality factor from the integrated passive components [8]. Therefore, additional research on PA modeling is necessary to ensure the precise design of digital or analog pre-distorters, envelope tracking, and linear Doherty PAs [9]. Furthermore, harmonically tuned and switch-mode power amplifiers are widely investigated, since it is proven that they can achieve higher efficiencies compared to the conventional linear PAs [10].

This paper includes an extended analysis of the design flow that was followed for the implementation of a 37–40 GHz SiGe BiCMOS quasi-inverse class F power amplifier presented in [11]. The key differences and main contributions of the present paper are summarized as follows: in contrast to [11], a comprehensive theoretical description of the proposed PA class is provided in the present work, as well as post layout information for some key passive parts of the amplifier. Moreover, new experimental results have been added, enforcing the understanding of the proposed PA class and providing information about the response and characteristics of the fabricated PA. Moreover, the PA's response to modulated-signal excitation has been examined after the creation of a simplified PA model.

The present work is organized as follows: the ideal quasi-inverse class F power amplifier and its comparison with the conventional inverse class F PA are described in Section 2. An analytical model for the main active core of the fabricated Ka-band PA along with a novel technique for the cancellation of the output parasitic capacitance of our active device are presented in Section 3. Afterwards, Section 4 provides a concise overview of each part of the proposed PA, in both a schematic and physical level. In Section 5, a straightforward model is presented for plotting the voltage and current waveforms after the parasitic output capacitance has been extracted. The extraction of this capacitance is a critical step in the process, and the model provides a simplified way to visualize the resulting waveforms. Section 6 denotes both the simulation and experimental validation that were conducted to assess the effectiveness and reliability of the proposed PA. Finally, in Section 7, the conclusions of the present work are drawn.

## 2. Ideal Quasi-$F^{-1}$ Power Amplifier

A conventional inverse class F PA is defined as ideal if the ideal harmonic-impedance conditions are realized at the output node of the active device under test. More specifically, not only a proper transistor's biasing is required, but also an infinite number of odd-harmonic and even-harmonic tank resonators are necessary to be included in the output network of an ideal inverse class F PA. In such a case, control of an infinite number of even and odd harmonics are realized, resulting in zero overlapping of the square current and half-sinusoidal voltage waveforms across the active device [12]. The latter impose 100% efficiency and, thus, zero power dissipation. However, it is noteworthy to highlight that only a finite number of harmonics will be present at the transistor's load in practice. The above restriction comes from the band-limiting behavior of the passive output network as well as the chip area limitations [13]. As follows, nonideal square current and half-sinusoidal voltage waveforms are present across the active device under test and, thus, power proportional to the V-I overlap is dissipated. An interesting solution to enhance the power profile of higher harmonics is presented in [14], where cascaded p-n junctions for carrier injection engineering with poly-silicon are used. Table 1 summarizes the theoretical maximum efficiencies that can be achieved for various combinations of voltage and current harmonic components that are present at transistor's load. The analysis and calculation of the presented efficiencies can be found in [10]. Due to the restrictions mentioned above, most of the inverse class F PAs realized in practice limit the harmonics control up to the third of a fundamental frequency, setting the upper limit of their maximum efficiency at 75% [13,15–17].

**Table 1.** Resultant efficiencies for various combinations of voltage and current harmonic components [10].

| Voltage Harmonic Components | Current Harmonic Components | | | |
|---|---|---|---|---|
| | 1 | 1, 3 | 1, 3, 5 | 1, 3, 5, $\ldots$, $\infty$ |
| 1 | 0.500 | 0.563 | 0.586 | 0.637 |
| 1, 2 | 0.667 | 0.750 | 0.781 | 0.849 |
| 1, 2, 4 | 0.711 | 0.800 | 0.833 | 0.905 |
| 1, 2, 4, $\infty$ | 0.785 | 0.884 | 0.920 | 1 |

A quasi-inverse class F power amplifier imitates by half the behavior of a conventional inverse class F power amplifier concerning the harmonically tuned output termination. The distinctive feature of the proposed PA class is its ability to regulate only the second harmonic component of the voltage output across the active device, which streamlines the design process for the output-matching network. Unlike conventional inverse class F power amplifiers, the proposed PA class does not utilize a third harmonic resonator in the output current, avoiding the insertion of additional losses during high-frequency operation. The limited quality factor (Q) of passive components frequently causes the third harmonic resonator to cancel the expected improvement in efficiency. According to Table 1, the upper limit of the proposed quasi-inverse class F is 66.7%. However, this efficiency limit is reduced even more, as it happens in the case of the conventional inverse class F PA, due to finite knee and breakdown voltages of the active device as well as the resistive losses of the passive load network.

### 3. Modelling of the Active Device

To achieve an accurate modeling of a mm-wave PA, it is crucial to thoroughly examine the main amplifying core or active module of each stage, as well as the electromagnetic (EM) characterization of its passive structures. By considering the active module as a voltage-dependent current source with an input and output admittance $Y_{in} = G_{in} + jB_{in}$ and $Y_{out} = G_{out} + jB_{out}$, respectively (Figure 1a) [18], a "loop gain" analysis can be performed at node X not only to assess stability, but also to estimate the output admittance. This analysis involves "breaking" the circuit at the desired node while still maintaining overall performance. The result of the described process can provide insight into the output admittance ($Y_{out}$). Such an analysis is beneficial in the circuit design flow, especially for multiple-stage PAs with or without feedback loops (Figure 1b) compared to the conventional large-signal S-parameter analysis (LSSP), which is not able to provide information regarding the input and output impedances of the intermediate stages, since it calculates the coefficients related only to the defined terminations.

The amplifying core model of the proposed single-stage PA is depicted in Figure 1c, and it incorporates the cascode topology as a single active device (AD). By separating the AD from the output-matching network (OMN), the output capacitance ($C_{out}$) can be determined as:

$$C_{out} = B_{out}/\omega \tag{1}$$

where $\omega$ is the angular frequency. The presented ideal RF choke provides a transparent dc current path.

As follows, we can determine the input and output impedances of the device under test while the input power is being varied. For that purpose, we can perform a harmonic balance simulation in which the proposed nonlinear circuit is represented as a superposition of harmonic components and the steady-state behavior of each harmonic component is analyzed separately. Once the steady-state behavior of each harmonic component has been determined, the output admittance can be calculated by dividing the harmonic voltage across the desired nodes by the corresponding harmonic current at each harmonic frequency.

This allows us to accurately capture the changes in the device's output capacitance in relation to its driving power, thereby enhancing the precision of the design process.

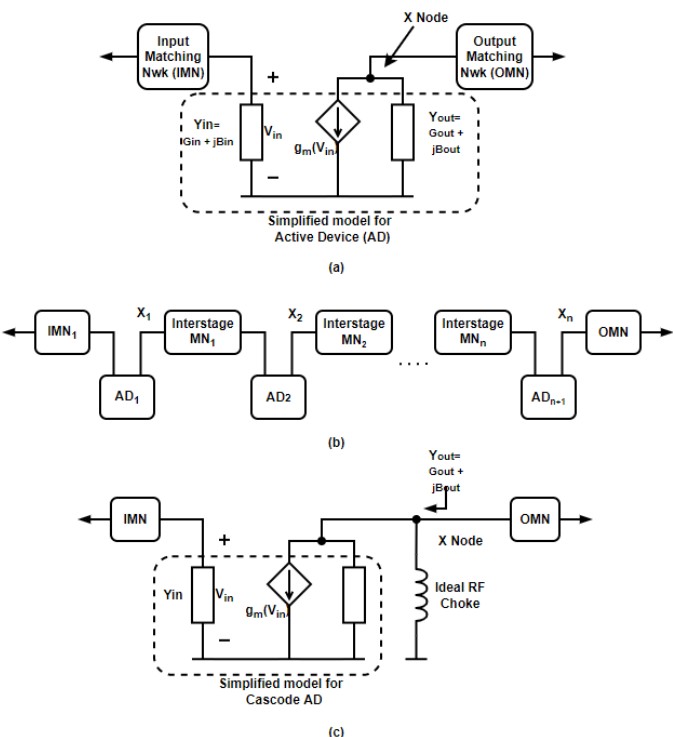

**Figure 1.** (**a**) Simplified model for a PA's active device and X node notation. (**b**) X nodes of a multiple-stage PA. (**c**) Simplified model for the cascode active device of the proposed PA.

The results of the harmonic balance "loop gain" analysis, performed over a frequency range of 37–40 GHz, are presented in Figure 2. The figure displays the variation in the output capacitance $C_{out}$ of our active device with respect to the input power ($P_{in}$) sweep. It should be mentioned that the active device under test includes all the layout parasitics imposed by transistors' via and the interconnects between the common-emitter (CE) and common-base (CB) HBT of the cascode structure. To ensure stability at the desired frequency band, ideal conjugate matching is employed at the input of the active device and a 15 $\Omega$ series resistor is inserted to the base of the CE HBT $Q_1$. The results indicate that $C_{out}$ is approximately 115 fF at low input power, while, at the operating point where the power gain is 1-dB compressed ($\sim$ 4–5 dBm), $C_{out}$ is found to be in the range of 120–125 fF.

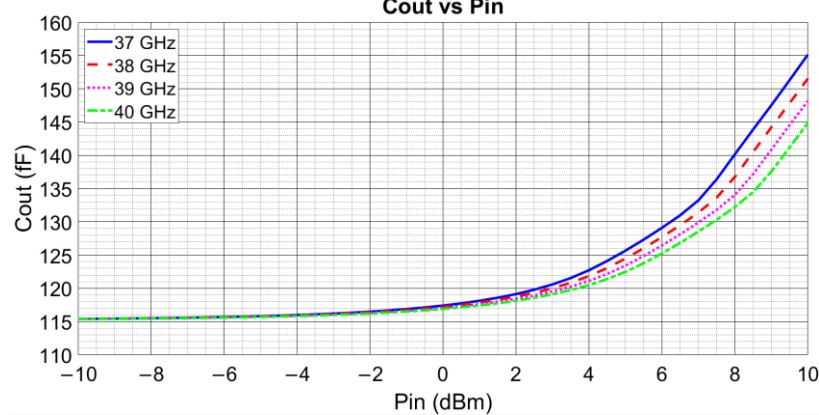

**Figure 2.** Output capacitance $C_{out}$ vs. the available input power $P_{in}$.

The validity of the above results can be straightforwardly confirmed by verifying that the extracted output capacitance is effectively cancelled out by the required inductance [19]:

$$L_{req} = \frac{1}{\omega^2 C_{out}} \tag{2}$$

By conducting a swept power load pull analysis at 38 GHz, it was found that the power contours at the 1dB compression point have a center located at impedance $Z_{Load} = 13.9 + j22.4\ \Omega$. However, after replacing the rf choke component with an ideal inductance of approximately 146 pH, the load pull contours are centered around the real axis at the optimum load $Z_{Load}' = 50 + j0\ \Omega$, as depicted in Figure 3 and discussed in [8]. This change in load impedance can be mathematically explained using the established equations for converting series to parallel impedance:

$$\begin{cases} Z_{Load} = R_S + jX_S \\ Y_{Load} = \dfrac{1}{R_P} - j\dfrac{1}{X_P} \xrightarrow{\text{with Cout canc.}} Y_{Load}' = \dfrac{1}{R_P} \\ Q = \dfrac{X_S}{R_S} = \dfrac{R_P}{X_P} \end{cases}$$

$$\Leftrightarrow Z_{Load}' = \frac{1}{Y_{Load}'} + j0 = R_P = \left(Q^2 + 1\right) \cdot R_S \tag{6}$$

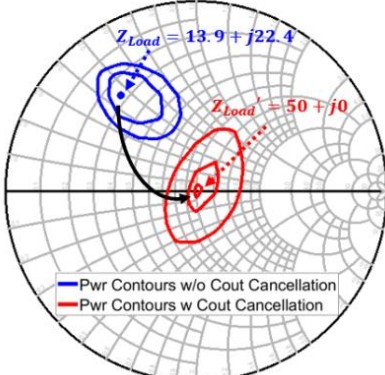

**Figure 3.** Power contours with and without $C_{out}$ cancellation. The power step is 0.5 dBm.

Considering the load Pull analysis as a benchmark for validating the discovery process of $C_{out}$, it is important to highlight that the proposed method based on the loop gain analysis of a single harmonic balance (HB) simulation has been found to produce more accurate results in comparison to the conventional large-signal s-parameter (LSSP) simulation. This novel method for investigating $C_{out}$ provides the ability to create a highly accurate model of PA classes that are harmonically tuned and highly efficient, such as Class F, $F^{-1}$, and E, as referenced in [20].

## 4. Proposed Quasi-Inverse Class F Power Amplifier

Figure 4 depicts the proposed single-stage, second-harmonically tuned, 37–40 GHz, quasi-$F^{-1}$ power amplifier. The circuit features a cascode topology as the selected active device, a simple but effective cascode current mirror serving as a bias network, and a T-type input-matching network. Additionally, the PA schematic incorporates an output harmonically tuned load to achieve an optimal 50 $\Omega$ fundamental impedance and an open-circuit second harmonic load. Both the input and output ports are terminated with a 50 $\Omega$ impedance.

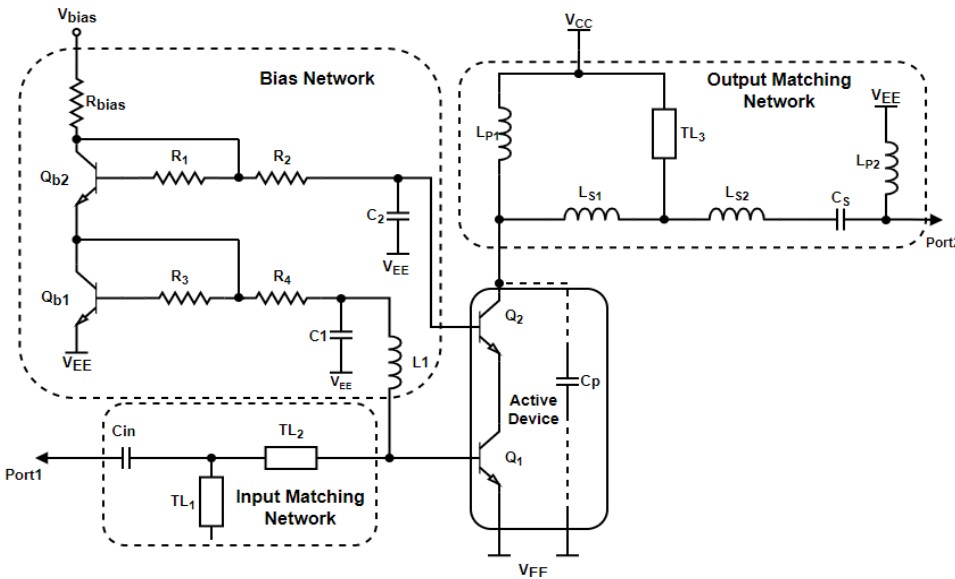

**Figure 4.** Schematic of the proposed single-stage quasi-F$^{-1}$ power amplifier.

The quasi-F$^{-1}$ PA was designed and fabricated in Infineon's 130 nm SiGe BiCMOS process. The process includes high-speed NPN HBTs with a unity gain frequency of $f_T = 250$ GHz and a maximum frequency of oscillation $f_{max} = 370$ GHz. The breakdown voltages of the provided HBTs are $BV_{CEO} = 1.8$ V and $BV_{CBO} = 5.3$ V. The layer stackup profile includes six copper metal layers and a 1.0 μm aluminum layer serving as the top metal. In the following subsections, each part of the circuit is analyzed in both schematic and physical layout levels.

It is worth mentioning that the design steps that followed for the present work form a robust methodology for designing and fabricating harmonically tuned PAs in various technology nodes for potential performance improvements. For instance, scaling down to a smaller technology node, such as 90 nm SiGe BiCMOS process, can offer improved response in terms of higher cutoff frequency, lower noise figure, and increased power density. This can translate into higher overall performance (output power and power gain) and efficiency for the PA. However, there are also some potential trade-offs to consider, such as increased device mismatch and increased power dissipation. On the one hand, device mismatch can be a significant issue when scaling down to a smaller technology node, as the smaller devices can exhibit greater variation in their electrical characteristics. This can lead to reduced gain, higher noise figure, and decreased linearity in the PA. On the other hand, smaller devices can lead to higher power density, which, in turn, can result in higher temperatures and increased power dissipation and reliability issues.

### 4.1. Supply Voltage $V_{cc}$

Aiming for a maximum output power at the saturation point of 20 dBm on a 50 Ω load, which is considered as the input impedance of SISO transmitter antenna, the required voltage amplitude at the fundamental is:

$$P_{sat} = \frac{V_{rms@fund}^2}{R_{Load}} \Leftrightarrow V_{fund} = \sqrt{P_{sat} \cdot R_{Load} \cdot 2} \approx 3.16 \text{ V} \tag{7}$$

According to [13], the maximum PA efficiency is achieved when the voltage amplitude at second harmonic is $V_{2nd} \approx 0.354 \cdot V_{fund}$ and is given by:

$$n_{max} = 0.816 \cdot \left(1 - \frac{V_{knee}}{V_{CC}}\right) \tag{8}$$

where $V_{knee}$ is the knee voltage of the active device and $V_{CC}$ the supply voltage. After taking a 700 mV knee voltage margin into consideration due to cascode configuration of our active device, Equation (8) implies that, in maximizing $V_{cc}$, the maximum efficiency increases too. Consequently, a 3.3 V supply voltage is selected, since, as it is imposed from the particular technology rules, it is the maximum allowable supply voltage for safe and reliable operation.

It should be noticed that the AD's collector will experience a maximum of approximately 7.2 V of sinusoidal peaking. This peak will not cause any issues, as the base node of the selected AD will be terminated with a finite resistance through a DC bias network, providing a path for discharging the base to ground and preventing base charge accumulation. This, in turn, will avoid early collector impact ionization. The breakdown voltage of the active device will realistically be limited by $BV_{CBO}$, which is exceptionally high in the case of a cascode topology composed of SiGe HBTs. Furthermore, the supply voltage is fed into our circuit via an aluminum pad, while diode circuits provide the required ESD protection.

*4.2. Cascode Topology*

The cascode topology has been selected as the main amplifying core for its numerous advantages. One of the key factors that motivated this choice was the higher gain and reduction in the Miller's effect, which enhances the stability of the active device, as highlighted in references [18,21].

The size of $Q_{1,2}$ has been carefully selected to deliver an output power of more than 18 dBm when biased in a deep class AB point, also known as the "sweet spot" [10]. This bias point allows the HBT to produce a maximum current at the first harmonic, similar to when it is biased in a class A point, while also reducing the maximum collector current of the third harmonic [18]. The bases of $Q_{1,2}$ are biased at approximately 0.81 V and 1.65 V, respectively, resulting in a cascode branch that conducts approximately 12 mA of quiescent collector current. The effective emitter length of the selected double-emitter HBTs (CBEBEBC) is $6 \, bl \cdot 2 \cdot 2.8 \, \mu m$, resulting in a current density of approximately 11.5 mA/$\mu m^2$ at maximum $f_T = 250$ GHz when the active device is driven close to 1 dB compression point ($OP_{1dB} = 18.6$ dBm). The described biasing leads to the AD conducting $I_{DC} = 52$ mA at 1 dB compression point, maintaining a power gain of around 16 dB at 38 GHz.

It is worth mentioning that the intrinsic interconnects of the HBTs can be proven crucial not only to efficiency, but also to the optimal impedance matching at the input and output of our AD. Thus, the via interconnects were EM simulated and their effects were taken into consideration for an accurate modeling of the HBTs.

*4.3. Bias Circuit*

The proposed bias network is depicted in Figure 4 with a dot line. It is a simple yet effective current mirror that consists of the transistors $Q_{b1}$, $Q_{b2}$, TaN resistors $R_{bias}, R_{1-4}$, MIM capacitors $C_{1,2}$, and a large inductor $L_{bias}$. The size of the included HBTs and resistors determine the dc voltages used for biasing of the cascode AD. More specifically, $R_{bias} = 875 \, \Omega$ while $\frac{R_{1,3}}{R_{2,4}} = \frac{l_{e, \, Q_{1,2}}}{l_{e, \, Q_{b1,b2}}} = 15$, generating the aforementioned dc voltages 0.81 V and 1.65 V at the bases of $Q_1$, $Q_2$, respectively. Furthermore, a large octagonal inductor $L_{bias}$ realized in top metal isolates the bias circuit from the active device and the rf path, while the decoupling capacitors $C_1 = 5$ pF and $C_2 = 1$ pF provide a low impedance path to the ground at the Ka-band.

It is noteworthy to highlight that a parasitic extraction process has been conducted on all elements contributing to the direct current path, including HBTs and resistors, while components contributing to the rf path have been EM simulated.

*4.4. Output-Matching Network*

The implementation of the proposed OMN is based on [15] and adopts a harmonically tuned circuitry, offering a transparent current path for the first harmonic as well as an effective short circuit of the second harmonic.

The identification of the parasitic capacitance $C_p$ of the active device is vital for determining the optimal values of the components that form the output network. This capacitance can be derived and authenticated through the techniques described in Section 2 with ease. Cancelling out the parasitic capacitance $C_p$ via the inductor $L_{p1}$, the optimum load impedance is transferred close to the real axis of the Smith Chart.

Upon the completion of the parasitic capacitance $C_p$ extraction process, the calculation of the optimal inductances and capacitances of the output-matching network becomes a matter of simple procedure. At the fundamental operating frequency $f_0$, the short-circuit termination located at one end of the quarter-wavelength line ($\lambda$/4) transforms into an open circuit at its opposite end. As a result, the OMN transforms into a series configuration consisting of the shunt $L_{p1}$-$C_p$ tank in conjunction with $L_s = L_{s1} + L_{s2}$ and $C_s$, thereby constituting a dual $f_0$ resonator (as depicted in Figure 5a). It is worth noting that the inductor $L_{p2}$ cancels out the parasitic capacitance $C_{PAD} = 60$ fF that is inherent to the output rf pad. Moreover, the inductor $L_{p1}$ resonates out the parasitic capacitance $C_p$, while the series resonator $L_s$-$C_s$ provides a clear pathway for the transmission of the fundamental component of the current from the generator device ($Q_1$, $Q_2$) to the 50 Ω load.

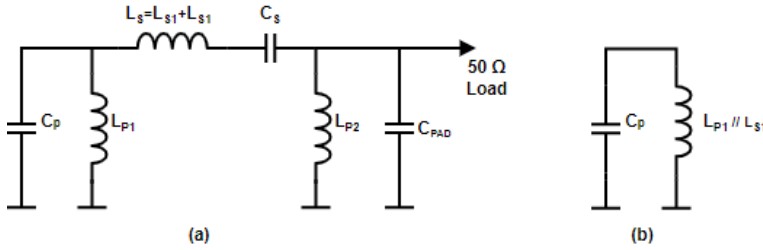

**Figure 5.** Schematic of the load network at (**a**) the fundamental $f_0$ and (**b**) the second harmonic.

At the second harmonic frequency, the quarter-wavelength line transforms into a short circuit, thereby disconnecting the PA from the output load. This transformation is effectively depicted in Figure 5b and results in the conversion of the load network into a parallel second harmonic resonator composed of $L_{p1}$ in conjunction with $L_{s1}$ and $C_p$, thereby providing the active device's transistors with an extremely high impedance ($Z_2 = \infty$) at the second harmonic. Table 2 provides a concise summary of the values of the passive components of the OMN utilized in this study, as well as the corresponding Q-factors at the fundamental. High Q-factors of the passive components are desirable in order to minimize the resistive losses of the output path and, thus, moderate the PA's efficiency degradation. For that purpose, T-line inductors constitute both $f_0$ and $2f_0$ resonators, since their quality factors are significantly higher (typical ~30–35) compared to on-chip spiral inductors (typical ~20–25).

**Table 2.** Passive values and Q-factors of the components of the output-matching network (OMN).

|  | $C_p$ | $L_{P1}$ | $L_{S1}$ | $L_{S2}$ | $C_S$ | $L_{P2}$ |
|---|---|---|---|---|---|---|
| Value @ $f_0$ | 120 fF | 146 pH | 49 pH | 97 pH | 120 fF | 290 pH |
| Q-factor @ $f_0$ | - | 32.2 | 36 | 34.8 | 17.5 | 26 |

Figure 6 demonstrates the 3D final layout view of the proposed output-matching network. In order to reduce the resistive losses and the capacitive coupling to ground, the top metal (M6) is used for the inductors and the signal path of the $\lambda/4$ transmission line, while the reference plane of the transmission line is realized in M4. Finally, the whole structure is EM simulated and phenomena such as capacitive coupling to ground, cross-talk, and mutual inductance have been taken into consideration.

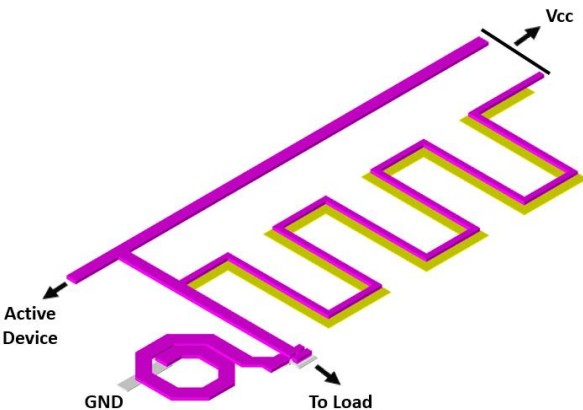

**Figure 6.** The 3D layout view of the proposed output-matching network.

*4.5. Input-Matching Network*

The designed input-matching network (IMN) shown in Figure 4 is a simple T-type network consisting of a 200 μm open stub and a 410 μm series transmission line, as well as an MIM capacitor $C_{in} = 110$ fF. The above matching network achieves a narrowband matching around 38 GHz between the source load and the main amplifier. The final 3D view of the proposed IMN is presented in Figure 7. As previously, the 50 Ω transmission lines are realized with M6 and M4 and the whole network is EM simulated.

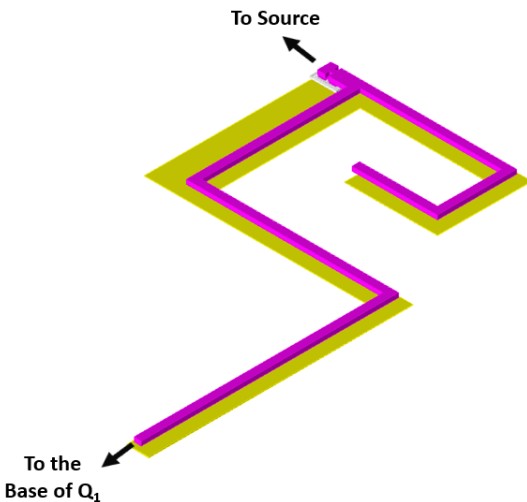

**Figure 7.** The 3D layout view of the proposed input matching network.

## 5. Intrinsic Voltage and Current Waveforms

The intrinsic collector current and voltage waveforms in the time domain, after the compensation of parasitic capacitance $C_{out}$, provide valuable insights into the performance of the active device when it is connected to a harmonically tuned load. The schematic model utilized to obtain the voltage and current waveforms at the output of the cascode topology is presented in Figure 8, as a direct access to the intrinsic current source node of the provided HBT model was not available. Our model consists of a multitone voltage source, which models the magnitude and phase components of the voltage (extracted from a harmonic balance analysis) at the collector node of the common base HBT $Q_2$ and an ideal capacitor, $C_{out}$, that represents the behavior of the AD's output parasitic capacitance over frequency. As $C_{out}$ changes by less than 4% at higher harmonics, a constant capacitance of $C_{out} = 120$ fF was selected to simplify the model.

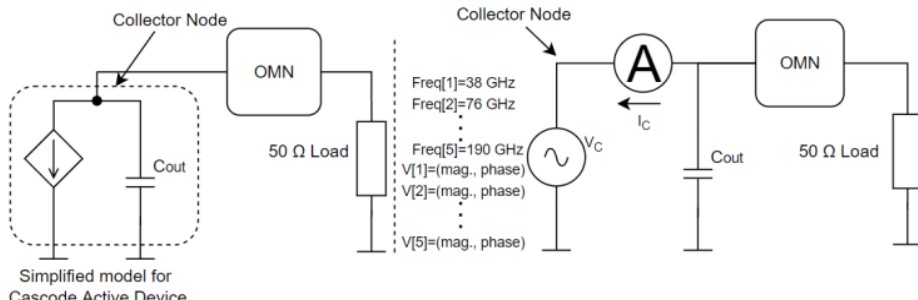

**Figure 8.** Simplified model for the extraction of the time-domain collector voltage and current waveforms.

An ammeter placed between the voltage source and the parallel combination of $C_{out}$ and OMN, measures the intrinsic current of our active topology. Moreover, in order to produce an accurate representation of the two waveforms, it is necessary to add the V—I DC components, as they have been extracted from a harmonic balance simulation of the proposed PA. It should be clarified that the active device constitutes the provided PDK transistor models along with the EM models that represent the metal interconnects for the performed harmonic balance simulation. The time-domain collector current and voltage waveforms at the 1 dB gain compression point operating at 38 GHz are displayed in Figure 9c and correspond to a PAE performance of approximately 40%. The strong presence of the second harmonic in the intrinsic collector voltage waveform (Figure 9a) results in a waveform that can be approximated as half-sinusoidal, while the weak and moderate presence of the second and third components of the current waveform results in an approximately square waveform (Figure 9b). These waveform shapes imply reduced V–I overlap and, therefore, improved efficiency.

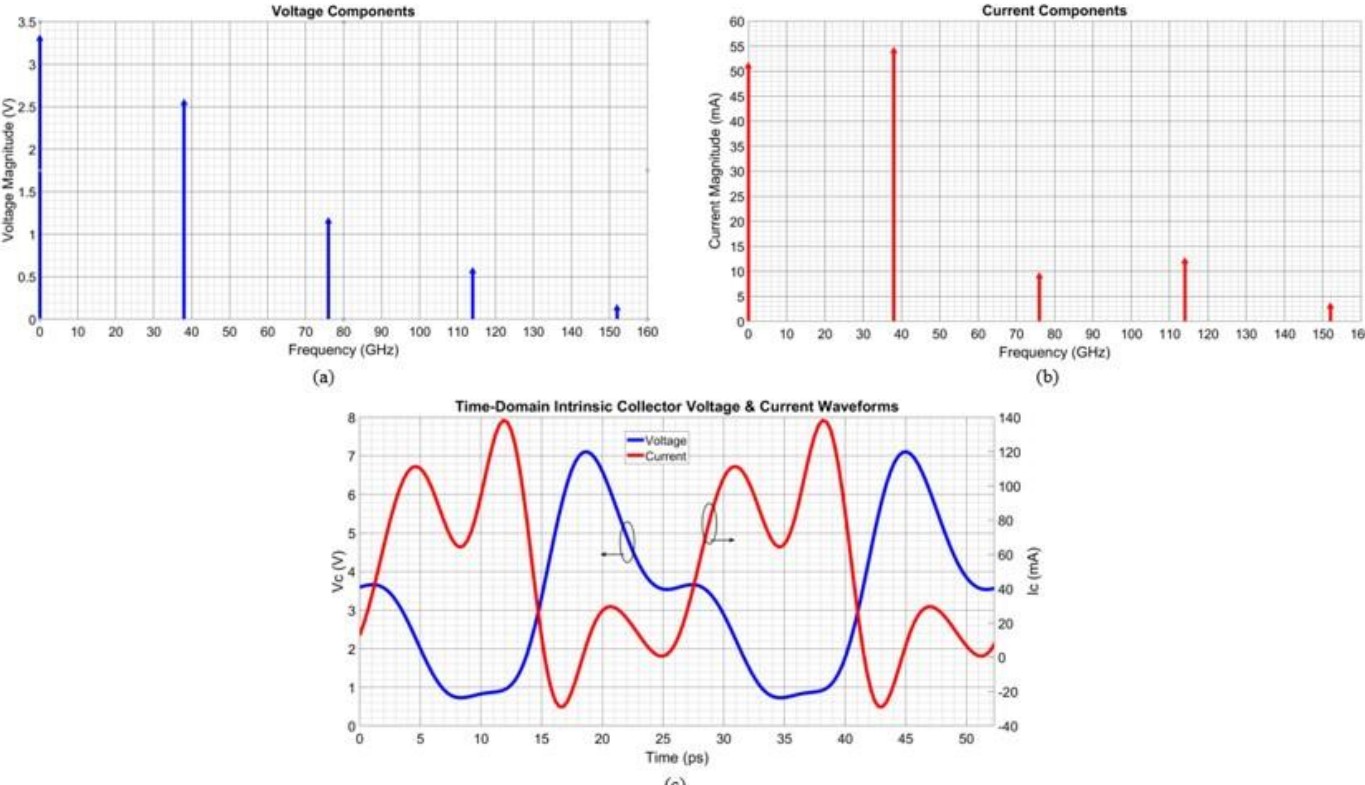

**Figure 9.** Simulated frequency-domain collector: (**a**) voltage and (**b**) current. (**c**) Time domain voltage and current waveforms.

## 6. Simulation and Measurement Results

Figure 10 depicts a photograph of the bare die chip. The total area of the fabricated chip is $0.605 \times 0.712$ mm$^2$, including all pads. In order to assess the performance of the designed quasi-inverse class F power amplifier, probe station measurements were conducted. A VNA analyzer was used for measuring PA's response during small-signal excitation, while the large-signal measurements were carried out using an rf signal generator and a power meter. It should be mentioned that the rf cable loss was thoroughly characterized over the frequency range of interest and any necessary adjustments were made to account for this loss through de-embedding.

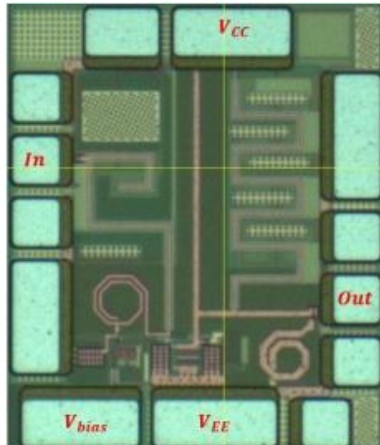

**Figure 10.** Bare die chip photograph.

Figure 11 displays the small-signal s-parameter measurement and simulation results of the proposed power amplifier. It should be noted that PA's dc operating conditions during small-signal measurement and simulations were a 3.3 V supply voltage ($V_{cc}$) as well as a 2.1 V bias voltage ($V_{bias}$). The above dc conditions result in almost 35 mW power consumption under dc operation. Furthermore, the s-parameter results indicated that the measured $S_{21}$ parameter exceeded 13 dB over the frequency range of 37–40 GHz, reaching a peak value of 15.6 dB at 37 GHz, while the $S_{11}$ parameter remained less than $-10$ dB over the frequency range of 38–40 GHz. Further broadening of input reflection coefficient is possible using more complex matching networks at the expense of area consumption. Moreover, Figure 11 depicts a moderate discrepancy between the $S_{22}$ measurement and simulation results over the frequency range 30–37 GHz. That inaccuracy is caused by calibration error due to the defective calibration substrate used in the s-parameter measurement. However, the aforementioned difference between the $S_{22}$ measurement and simulation results becomes negligible over the frequency range of our interest (37–40 GHz). Additionally, the PA demonstrates stability throughout the entire frequency range, as depicted in Figure 12 through the k-factor measurement.

Regarding the large-signal measurement, as depicted in Figure 13, the proposed quasi-inverse class F power amplifier demonstrates remarkable performance, with a large-signal power gain of more than 14 dB at its operating frequency of 38 GHz. Additionally, the output 1 dB compression point, represented by $OP_{1dB}$, is approximately 17.6 dBm, while the saturated output power $P_{sat}$ is around 19 dBm and the maximum $PAE$ reaches 33%. It should be noted that the linear response of the proposed Ka-band PA is essential for maintaining high signal fidelity and minimizing interference, distortion, or nonlinear effects in wireless communication systems. Further improvement of the linearity of the presented PA is possible if a linearization technique is applied. Some of the commonly used linearization techniques in mm-wave PAs include predistortion, feedback, feedforward, digital predistortion, and envelope tracking [18].

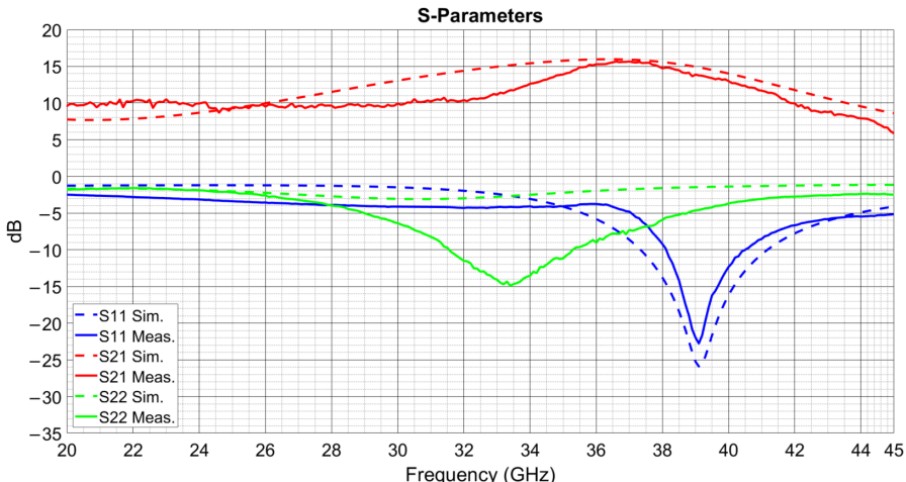

**Figure 11.** Small-signal s-parameters measurement and simulation results.

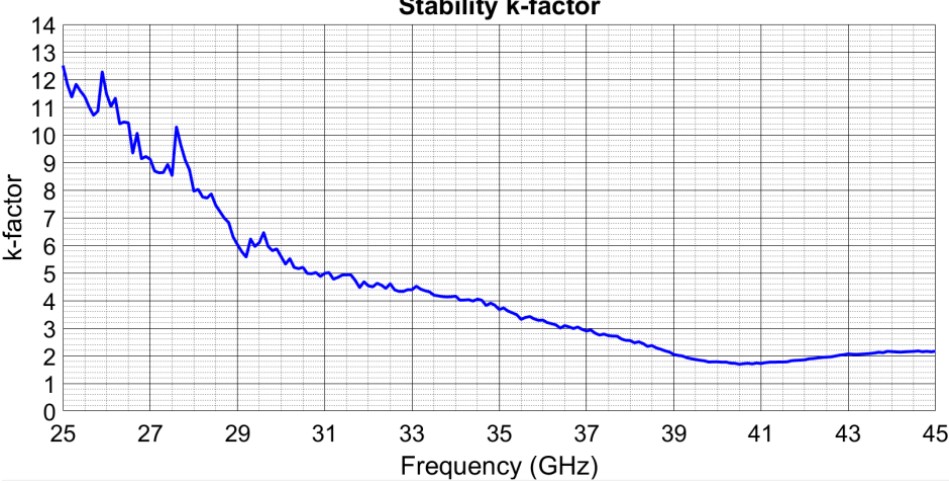

**Figure 12.** Stability k-factor measurement.

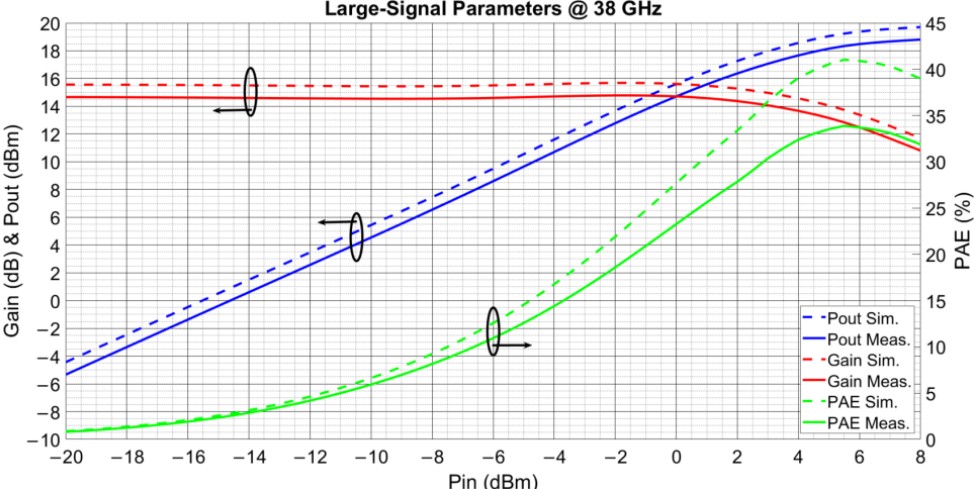

**Figure 13.** Large-signal measurement and simulation results at 38 GHz.

Examining Figure 14, which plots the $OP_{1dB}$ and $PAE$ over the frequency range of 37–40 GHz, it becomes evident that the PA maintains high performance, with $OP_{1dB}$ greater than 16 dBm and $PAE$ higher than 30% throughout the entire band of interest. Other

crucial specifications determined through our measurements include the $IP_{1dB}$, which is approximately 9 dBm, and the AM-to-PM conversion, which is less than 13° for all the swept input powers (−20~8 dBm). The AM-to-PM conversion holds its maximum of 13° when the proposed PA operates in saturation. Efficiency (%) at 3 dB and 6 dB power back-off have been extracted from the results of the performed large-signal measurements at 38 GHz and they are highlighted in Figure 15. In particular, the designed quasi-inverse class F power amplifier achieves a 3 dB back-off collector's efficiency of around 27.8% and a 6 dB back-off efficiency of 19.3%. Finally, the dc current drawn by our active device versus the output power deliver to the load is plotted in Figure 16. It should be mentioned that the dc current consumption under no input excitation is around 10 mA ($P_{dc} = 33$ mW), while the designed PA draws 52 mA dc current when the output power reaches 1 dB compression point. Further improvement of the quiescent power consumption $P_{dc}$ is possible by reducing the bias voltage $V_{bias}$ of the common-emitter HBT driving our active module close to a class B operating point. However, such a modification leads to a reduction in the achievable power gain.

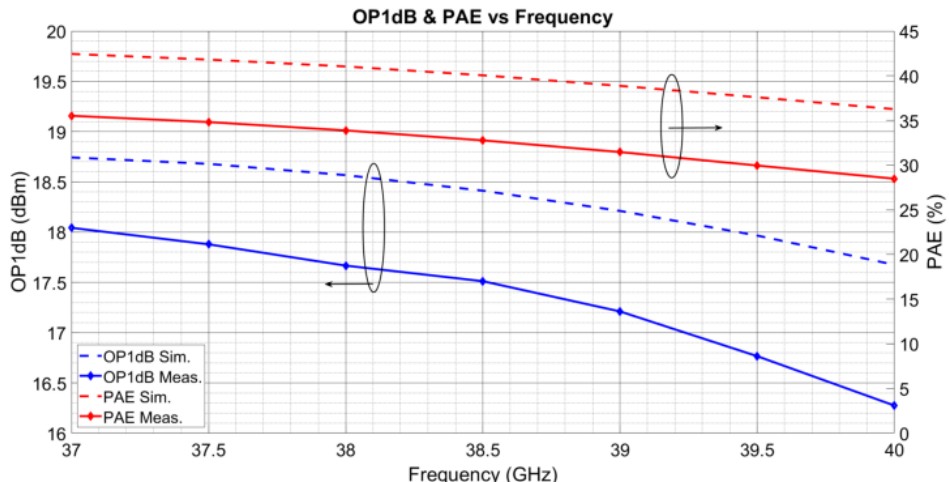

**Figure 14.** Measurement and simulation results of $OP_{1dB}$ and $PAE$ over frequency.

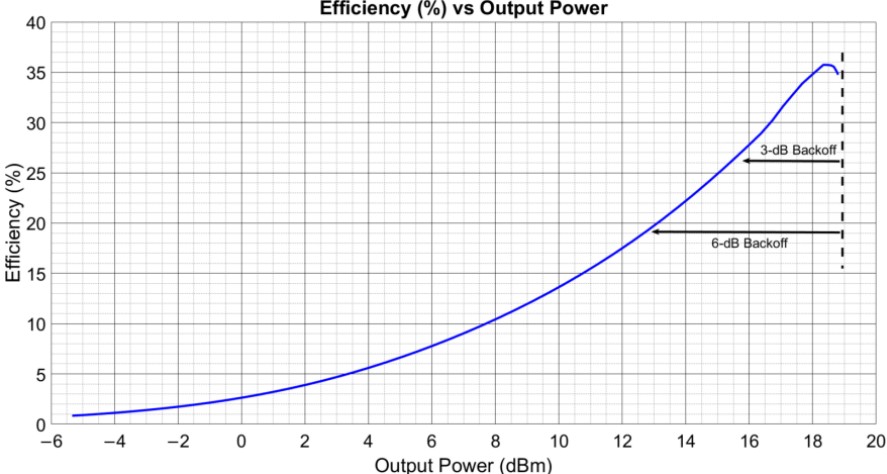

**Figure 15.** Measurement of PA's efficiency (%) vs. the output power. The 3 dB and 6 dB back-off efficiencies are denoted.

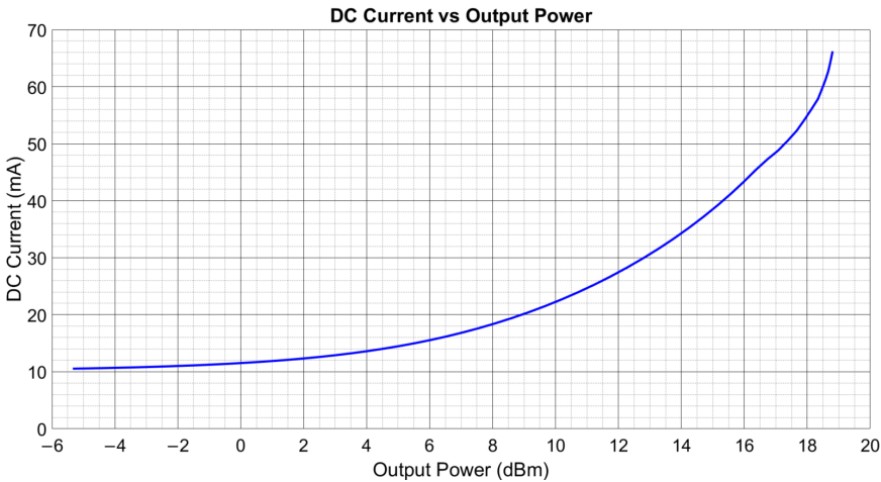

**Figure 16.** Measurement of the DC current (%) vs. output power.

### 6.1. Complex Waveform Simulations

Key top-level specifications that arise from the large-signal measurement results of the fabricated quasi-inverse class F power amplifier and have been presented above are incorporated in a simulation model in order to examine PA's response to modulated-signal excitation. The amplifier's model is created through Keysight's Pathwave System Design tool "SystemVue" that offers an advanced prototyping and design platform for complex RF systems. It is noteworthy to highlight that the following simulations and modeling require no extra RF instrumentation, forming an approximation based on the single-tone large-signal measurement results that have been already performed.

Keeping as reference the large-signal measurement at 38 GHz, the following entries have synthesized the amplifier's model:

- Saturation power: $P_{sat} = 18.8$ dBm;
- Output 1 dB compression point: $OP_{1dB} = 17.6$ dBm;
- Power gain: $G = 14.7$ dB;
- Gain compression at saturation: $GCSat = 4$ dB;
- Center frequency: $F = 38$ GHz;
- 3 dB bandwidth: $BW = 6$ GHz.

Concerning PA's excitation, a pseudo random binary sequence generator along with an arbitrary digital modulation source generate the modulated signals at a carrier frequency of 38 GHz. Keeping the input signal's bandwidth constant at 25 MHz in each test, an attempt is made to examine PA's response to QPSK and 16 QAM signals having different peak-to-average power ratios. It should be noted that, after a raised cosine filtering with 0.2 roll-off factor, the maximum available bandwidth is 30 MHz, while adjacent channel power ratio (ACPR) is measured at 27 MHz offset from the 38 GHz carrier frequency, as is depicted in Figure 17. The aforementioned simulation scenarios result in the bit rates of 50 Mbps and 200 Mbps for the QPSK and 16 QAM signals, respectively.

As is highlighted in Figure 18, the increase in average power efficiency entails increase in the error vector magnitude metric. Moreover, due to the fact that QPSK is the least PAPR modulation format, the PA's average power efficiency is slightly higher for a given EVM. Referring to Figure 19, the resulting EVM at PA's 1 dB compression point ($OP_{1dB} = 17.6$ dBm) is around 6% for the QPSK signal and 8% for the 16 QAM signal, while, for output powers that correspond to the linear region of the designed PA, the EVM is lower than 4% for both modulation schemes.

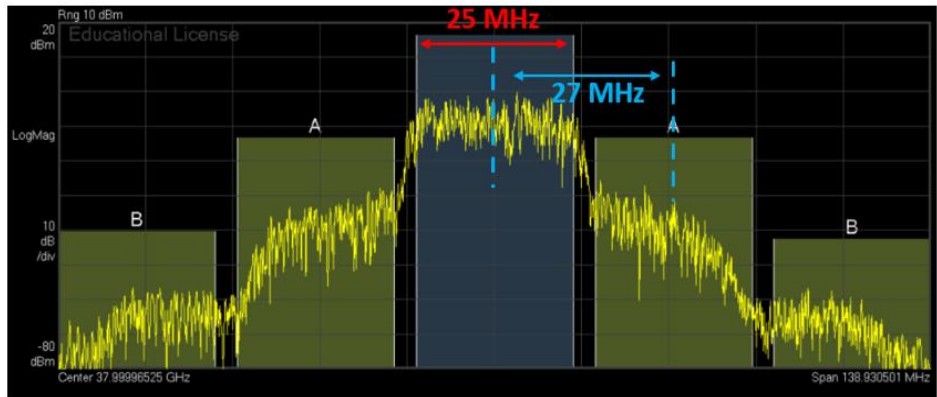

**Figure 17.** Signal bandwidth representation at the output of the PA model.

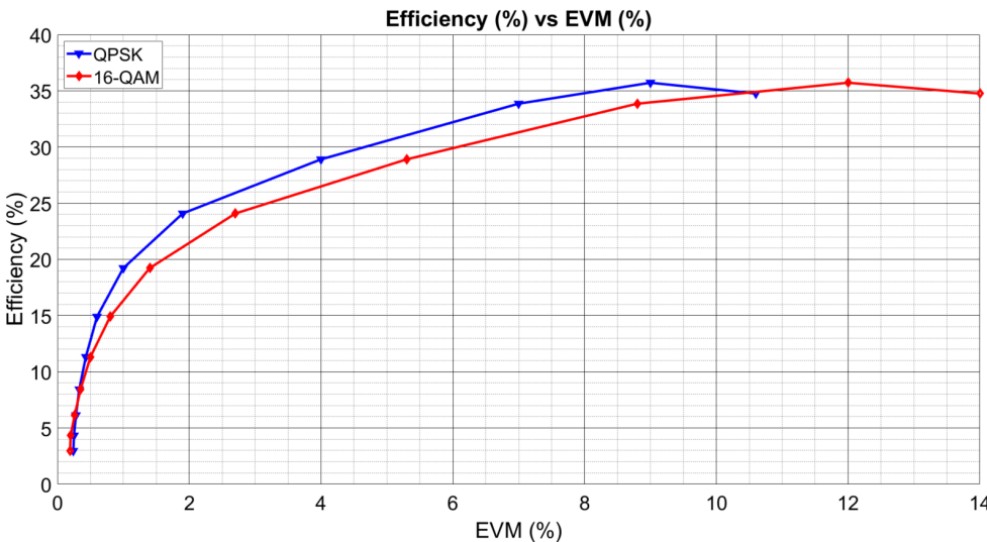

**Figure 18.** Efficiency (%) vs. EVM (%) for QPSK and 16 QAM signals with different peak-to-average power ratios.

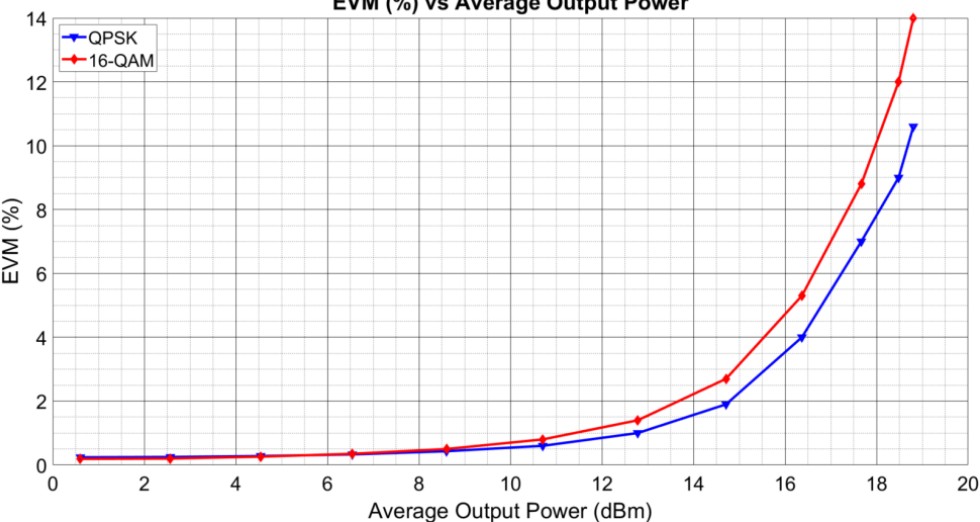

**Figure 19.** EVM (%) vs. average output power for QPSK and 16 QAM signals with different peak-to-average power ratios.

Based on the information provided in Figure 20, it can be observed that the adjacent channel power ratio (ACPR) exhibits a value of −40 dBc when the power amplifier (PA) model is stimulated with QPSK or 16 QAM signals, which have weak peak-to-average power ratios. It is important to note that, as the average output power of the PA increases, the ACPR also experiences an increase, which is a commonly expected phenomenon. When our PA reaches its 1 dB compression point, the ACPR is approximately −25 dBc for both input excitations. Finally, the ACPR requirement for 3GPP NR carrier [22] is depicted in Figure 20. The ACPR limit has been set at −26 dBc for the FR2 frequency range 37–52.6 GHz. As follows, the proposed PA model complies with the ACPR requirements of the 3GPP NR standards until its output almost reaches the $OP_{1dB}$.

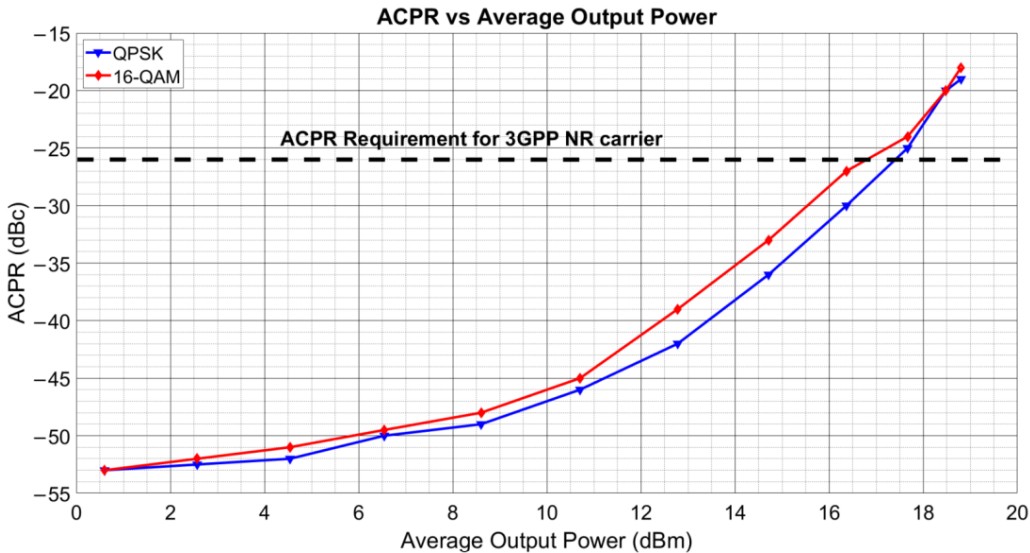

**Figure 20.** Adjacent channel power ratio vs. average output power for QPSK and 16 QAM signals with different peak-to-average power ratios.

*6.2. Comparison Table*

Table 3 includes a summary of the various parameters of the designed Ka-band quasi-inverse class F power amplifier, as they have been extracted from the performed measurements. Moreover, it provides the specifications of other state-of-the-art Ka-band power amplifiers integrated in SiGe BiCMOS technologies, making the comparison with the proposed one feasible. It is noteworthy to highlight that the fabricated PA presented in this paper exhibits a very good compromise between the maximum output power it can deliver to the load, the achievable power gain, as well as the maximum PAE. Further improvement of the PAE and the corresponding power consumption is possible if additional harmonic tanks are incorporated in the designed output-matching network for a higher order harmonic component control. The performance metrics of Table 3 demonstrate that the proposed PA based on the quasi-inverse class F technique is competitive to other high-efficiency techniques that adopt high-order multi-resonance harmonic filter loads. A Figure of Merit (FoM) has been introduced by the authors of this article for performance comparison of the various Ka-band SiGe PAs. The FoM is calculated using the following formula:

$$FoM = P_{sat}(\text{dBm}) + OP_{1dB}(\text{dBm}) + 10\log(PAE(\%)) + 20\log(Frequency(\text{GHz})) - 10\log(P_{dc}(\text{mW})) \qquad (9)$$

As is clearly depicted in Table 3, the present work achieves the highest FoM (68) among state-of-the-art integrated Ka-band SiGe PAs, verifying the effectiveness of the proposed design and highlighting its potential for future applications in high-frequency communication systems.

**Table 3.** Comparison table of published Ka-band PAs.

| References | This Work | [9] | [13] | [15] | [17] | [21] | [23] | [24] | [25] |
|---|---|---|---|---|---|---|---|---|---|
| Technology | 130 nm SiGe | 130 nm SiGe | 130 nm SiGe | 130 nm SiGe | 130 nm SiGe | 130 nm SiGe | 130 nm SiGe | 130 nm SiGe | 180 nm SiGe |
| $f_T/f_{max}$ | 250/370 | - | 180/220 | 180/220 | 200/265 | - | - | 180/220 | - |
| Class/Architecture | Quasi-$F^{-1}$ | Doherty | $F^{-1}$ | $F^{-1}$ | $F^{-1}$ | 4-way comb. | $F^{-1}$ | $F^{-1}$ | Digitally VGA |
| Stages | 1 | 2 | 2 | 2 | 2 | $2 \times 4$ | 2 | 1 | 2 |
| Frequency (GHz) | 37–40 | 37 | 38 | 39–42 | 38 | 35 | 28.5 | 31 | 25.8–35 |
| $P_{sat}$ (dBm) | 18.8 * | 17.1 | 16.5 | 18 ** | 21.2 | 22.8 | 17 | 17.1 | 11.1 *** |
| Gain (dB) | 14.7 * | 17.1 | 16.5 | 18 ** | 22.1 | 25.3 | 20 | 10.3 | 16 *** |
| $OP_{1dB}$ (dBm) | 17.6 * | 15.5 | 15 | 16 ** | 17.5 | 22.6 | 15.2 | 15 | 9.6 *** |
| PAE (%) | 33 * | 22.6 | 38.5 | 43 ** | 30.1 | 27 | 43.5 | 40.7 | 55.9 *** |
| $P_{dc}$ (mW) | 33 | 24.9 | 25.2 | 31.5 | 75 | 198 | 30.4 | 19.8 | 22.5 |
| Size (mm$^2$) | 0.43 ($^+$ 0.2) | 1.76 | 0.51 | 0.57 | $^+$ 0.76 | $^+$ 0.48 | $^+$ 0.29 | 0.27 ($^+$ 0.14) | 0.72 ($^+$ 0.23) |
| FoM | 68 | 63.5 | 64.9 | 67.4 | 66.3 | 67.6 | 62.9 | 65 | 54 |

* at 38 GHz. ** at 40.5 GHz, *** at 29.5 GHz, $^+$ size excluding pads.

## 7. Conclusions

In this paper, a SiGe BiCMOS power amplifier is successfully implemented and demonstrated at the Ka-band. The designed PA is based on a quasi-inverse class F technique that adopts a second-harmonically tuned load providing the required impedances to the cascode amplifying core. This paper fully discusses the potential and limitations of the proposed quasi-inverse class F technique, as well as its distinctive feature from the conventional inverse class F technique. In order for the quasi-inverse class F approach to be implemented, it is crucial to thoroughly examine and calculate the parasitic components of the PA's main active device. In this work, the cascode configuration is treated as a single active device, with its output capacitance being identified as a critical parasitic component. A detailed methodology is described for the discovery of the active device's output parasitic capacitance enforcing the accuracy of the main core modeling and enabling the designing of the harmonically tuned load. Furthermore, a comprehensive description of the design steps that were followed for the schematic and physical design of each part that constitute the proposed PA is presented. Afterwards, a simplified model for the extraction of time-domain intrinsic voltage and current waveforms is introduced, enforcing the process for the implementation of the proposed quasi-inverse class F technique. The major top-level specifications that have been identified based on the large-signal measurement outcomes of the quasi-inverse class F PA are included in a simulation model. This model is utilized to analyze the performance of PA when it is subjected to modulated-signal excitation. According to small- and large-signal measurements and the comparison table that includes other state-of-the-art Ka-band PAs, the designed single-stage amplifier achieves one of the highest output powers while maintaining a high level of efficiency.

**Author Contributions:** Conceptualization, V.M. and I.P.; methodology, V.M.; validation, V.M. and I.P.; investigation, V.M.; writing—original draft preparation, V.M.; writing—review and editing, V.M. and I.P.; visualization, V.M.; supervision, I.P. All authors have read and agreed to the published version of the manuscript.

**Funding:** This research received no external funding.

**Institutional Review Board Statement:** Not applicable.

**Informed Consent Statement:** Not applicable.

**Data Availability Statement:** Not applicable.

**Acknowledgments:** The authors would like to thank Infineon Technologies AG for fabricating the presented device as well as for their full support during the course of this work.

**Conflicts of Interest:** The authors declare no conflict of interest.

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
