# Peer review of "A Ka-Band SiGe BiCMOS Quasi-F−1 Power Amplifier Using a Parasitic Capacitance Cancellation Technique †"

_jlpea, doi:10.3390/jlpea13020023_

Round 1

Reviewer 1 Report

This work is an extension of a conference paper published in [11]. It is devoted to an issue of current interest. However, there are some technical concerns and incomplete information that must be addressed before publication.

1.          The work done in this paper is quite similar to the work reported in the conference paper [11] published by the same authors in the 2022 Panhellenic Conference on Electronics & Telecommunications. Specifically, both papers present circuit operation, theoretical analysis, simulation and measured results, parasitic capacitance cancellation technique, and simplified model of the proposed PA.

Could the authors highlight the key differences and main contributions of this paper?

2.          What are the limitations of the proposed model in Figure 1? Is the proposed model scalable with respect to devices’ geometry, temperature, biasing voltages, etc?

3.          To include the Q factor of the inductors shown in Figure 4. How does the Q factor of LS1 and LS2 affect the overall performance of the PA?

4.          Please include the power consumption of existing and proposed PAs in Table 3.

5.          How does the proposed PA compare with the following SiGe PA? The authors can include the comparison in Table 3 for easy reference.

K. Ma, et. al., "A Reconfigurable K-/Ka-Band Power Amplifier with High PAE in 0.18 µm SiGe BiCMOS for Multi-Band Applications," IEEE Transactions on Microwave Theory and Techniques (TMTT), vol. 63, no. 12, pp. 4395-4405, Dec. 2015.

6.          Please include a Figure of Merit (FOM) in Table 3 for performance comparison of the various SiGe PAs.

7.          What is the output power variation of your proposed PA design?

8.          Please comment on the performance and robustness of the proposed PA when smaller technology nodes such as 90nm SiGe BiCMOS process is used. What are the trade-offs?

9.          Based on the proposed PA as shown in Figure 4, it appears to me that one of the drawbacks of the proposed work is PDC, which makes it unattractive for low-power transceiver design. Please give some suggestions to reduce PDC.

10.      Keeping the operating frequency at 38GHz, what would be the peak power-added efficiency (???), saturation output power (????), and maximum large-signal power gain (G) when a 180nm SiGe BiCMOS technology is used instead?

11.      Figure 11. Please explain why the S22 measured results are so much better than the S22 simulation results.

Reviewer 2 Report

The paper is clear and interesting. There are however some important issues that need to be addressed:

1.     In section 3, how the harmonic balance simulation is used to calculate the output admittance should be explained in more detail, strongly supporting the novelty. In addition, it is said that the LSSP analysis cannot provide this information. I do not agree with this, as usually this is the method followed to design well-performing mm-wave PAs. Please provide some comparison or information to support this claim.

2.     The active device model looks oversimplified, given the high operation frequency. Please discuss why you are not including other parasitics such as for instance the CB capacitance.

3.     Page 4, line 139. Is the 15-Ohm resistor connected in series to the base? Please make it clear, saying for instance “… a 15 W series resistor is inserted to the base…”

4.     How are the simulations for Fig. 2 performed? Is it just with the PDK model, or also including layout parasitics?

5.     Once again, looking at Fig.3 and the discussion at the end of Section 3, I do not see the novelty of this approach. What is the difference with respect to using a LSSP or HBSP simulation? Another option would be to sweep the value of the collector inductor until it cancels the imaginary part of the Zopt, and extract Cout from there. Please discuss.

6.     Please revise the equation in line 188 of page 6. It is not clear how Vknee increases the peak voltage. 

7.     Please revise a typo in page 6, line 210. It should say “…resulting in a current density…”

8.     It is not clear why you use the simplified model in Figure 8 for the time domain analysis, instead of the PDK transistor model + EM simulation, which should account more for all the effects. 

9.     There is a strong discrepancy between the simulated and measured S22 parameters in Fig. 11. Please discuss this, especially because the output impedance extraction is one of the biggest claimed contributions of the paper.

10.  At which Pin is the AM-PM distortion of 20º given?

11.  Please provide more information about the amplifier model employed in the Pathwave simulations. For instance: is it possible to plot the Pout vs. Pin curve of that model and the measured one, to compare how accurate the model captures the non-linear behavior? Does it account for AM-PM?

12.  Please draw the spurious emission spectral mask in Fig. 17, to prove if the presented PA is suitable for 5G applications.

13.  Please make a line in the ACPR plot of Fig.20, to show the maximum value allowed by the standard.

14.  Looking at Table 3, it is not clear how the presented PA improves the state of the art. As it includes fewer matching structures, could the size be an advantage? In that case, please include the circuit area in the comparison.

Reviewer 3 Report

see attached file

Round 2

Reviewer 2 Report

Thank you for this revision. Most of my previous comments have been answered. I just have the following concern:

- Please mention if the FOM employed for the comparison with the state of the art has been taken from somewhere else. In that case, please reference. Otherwise, I would suggest using some of the typical PA FOMs used in the literature. If the authors "invented" the used one and continue using it, the I suggest including the operation frequency in the calculation. This is because usually some metrics like the Pout are more difficult to fulfill at higher frequencies.

Author Response

Dear Dr./ Mr. / Ms. Reviewer

Thank you very much for your response and comments. We really appreciate the effort that you have dedicated to providing your valuable feedback on our manuscript. We have been able to incorporate changes to reflect most of your suggestions. Below you can find our response to your comment.

Thank you for this suggestion. We have incorporated the operating frequency in the calculation of the FoM (See Page number: 16, Section: 6, Sub-section: 6.2, Line: 516-519). Also, we have included a comment in line 517 in order to make it clear that the authors of the present article have created this particular FoM  (See Page number: 16, Section: 6, Sub-section: 6.2, Line: 517).

Reviewer 3 Report

No further comments

Author Response

Dear Dr./ Mr. / Ms. Reviewer

Thank you very much for your response.